# Exploring the genetic diversity and population structure of aerial yams (*Dioscorea bulbifera L.*) DArT-seq and agronomic traits

Eunice Ekaette[1,2], Emeka Nwofia[1], Peter Okocha[1], Ikenna Nnabue[3], Kenneth Eluwa[3], Jude Obidiegwu[3], Paterne A. Agre[4]*

1 Department of Agronomy, Michael Okpara University of Agriculture, Umudike, Abia, Nigeria, 2 National Biotechnology Development Agency, Lugbe, Abuja, Nigeria, 3 National Root Crops Research Institute, Umudike, Abia State, Nigeria, 4 International Institute of Tropical Agriculture (IITA), Ibadan, Nigeria

* p.agre@cgiar.org

**Data Availability Statement:** All data can be downloaded from the yam base using the following link www.yambase.org.

## Abstract

*Dioscorea bulbifera* is an edible yam specie with aerial bulbils. Assessing the genetic diversity of *D. bulbifera* accession for cultivation and breeding purposes is essential for it genetic improvement, especially where the crop faces minimal attention. The aims of this study was to assess the genetic diversity of *Dioscorea bulbifera* accessions collected from Nigeria and accessions maintained at the genebank of International Institute of Tropical Agriculture (IITA) Ibadan. Accessions were profiled using quatitative and qualitative phenotypic traits and Diversity Array Technology SNP-markers. Multivariate analysis based phenotypic traits revealed high variability among the evaluated accessions and all phenotypic traits assessed were useful in discriminating the aerial yam accessions. Clustering analysis based phenotypic traits revealed the presence of two well defined clusters. Using DArT-Seq marker, the 94 accessions were classified into three genetic group through the admixture and the phylogeny analysis. The comparision of phenotypic and genotypic clustering revealed inconsistency membership across the two clustering methods. The study established a baseline for the selection of parental lines from the genetic groups for genetic improvement of the *D. bulbifera*.

## Introduction

Yams (*Dioscorea* L.) belong to the family *Dioscoreaceae*. *Dioscorea* L. is the largest genus of the family *Dioscoreaceae* [1, 2]. Yams (Dioscorea spp.) are popular staples in West Africa [3], serving as important sources of dietary calories. They contributed, on average, more than 200 kilocalories per person per day to over 300 million people between 2006 and 2010 [4]. The crop is widely adapted to several agro-ecologies in Nigeria, along with the ability of farmers to harvest their tubers at their convenience, makes it an essential food security crop [5]. For millions of people, yam tubers form an integral part of social-cultural activities [6] and medicinal values [7]. Nigeria ranks as the leading producer of yams (*Dioscorea* spp.) in the world, accounting for 65% (about 50.1 million tons) of annual global production [8]. Though yams are grown for

**Funding:** The funding support from core budget of Yam breeding program from the BMGF (OPP1052998), National Root Crops Research Institute (NRCRI) Umudike, Nigeria is acknowledged. The funders had no role in study design, data collection and analysis, decision to publish, or preparation of the manuscript.

**Competing interests:** The authors have declared that no competing interests exist.

their carbohydrate content, their storage organs are important sources of vitamins for millions of people in West Africa, Southeast Asia, and the Caribbean [9–11]. Many species also contain substantial amounts of vitamins such as carotene, thiamine, riboflavin, niacin and some minerals like calcium, phosphorus and iron [7, 12, 13]. There are over 600 species of yams world wide, of which about 12 species of economic significance as food plants [14, 15].

*Dioscorea bulbifera* (commonly known as the air potato, air yam) is native to Africa and Asia with Asian cultivars less angular, more spherical, and less toxic than African cultivars [5]. *D. bulbifera* has been therapeutic potentials in the management of ailments [7, 16]. The cultivation of *D. bulbifera* in Nigeria has remained traditional and is typically carried out by smallholder farmers. The lack of adequate information on genetic diversity limits its genetic improvement. To develop elite genotypes that combine high and stable yield of good tuber quality combine with diseases and pests resistant, a wide range of genetic diversity is required. Studies using agro-morphological do not demonstrate the true genetic relatedness of the accessions and are strongly influenced by environmental factors [17, 18]. In Nigeria, there is limited information on genetic diversity status of *D. bulbifera* using molecular markers. Molecular markers have been utilized as powerful tools for the estimation of genetic diversity of many species with great success and accuracy because they are abundant and unaffected by environmental parameters [19]. So far, markers used for assessing diversity in yams include restriction fragment length polymorphism (RFLP) of cpDNA [20], random amplified polymorphic DNA (RAPD) [21, 22] amplified fragment length polymorphism (AFLP) markers [23, 24], and SSR markers [25–28]. Next-generation sequencing (DArTseq-based) has been successfully applied in germplasm characterization in white guinea yam [29–32] water yam [33, 34], and in trifoliate yam [35], thus demonstrating its suitability for the high-throughput genotyping in yams. However, no genetic characterization study has been conducted on the *D. bulbifera*. This study aimed to better understanding of the genetic relationships and population structure of *D. bulbifera* in Nigeria. This will set the foundation for the crop improvement and development of breeding strategies in Nigeria.

## Materials and methods

### Experimental site

The experiment was conducted in the field during the 2020 and 2021 planting seasons at the eastern farm of the National Root Crops Research Institute, Umudike, located at 5˚20' N latitude and 07˚33'E longitude. The yam varieties were planted in the month of May while the harvesting was done in January. The annual rainfall ranges from 1800 to 2200 mm with an average annual air temperature and relative humidity of 20.50˚C and 76.80%, respectively. Umudike soil is characterized by a well-drainedred to yellowish sandy loam to sandy clays occuring at the summits and upper slopes of the ridges.

### Collection of planting materials

The planting materials used for this study consisted of 94 *D. bulbifera* accessions representing originated from different source. A total of 64 accessions were sourced from the genebank of International Institute of Tropical Agriculture (IITA) Ibadan, Nigeria. Thirty collections were also sourced from the National Root Crops Research Institute (NRCRI), Umudike, Nigeria (Table 1). The accessions were collected for research pupose only as agreed by the (NRCRI). The accessions were profiled for phenotypic traits and no extraits was tested on human. The study was conducted in the view of profiling D. bulbifera for breeding purpose. All activities was conducted in respect of the study design. All authors reviewed the manuscript and approved the submitted version.

**Table 1. List of *Dioscorea bulbifera* accessions, their origins and agro-ecological zones.**

| SN | Accession | Origin | Agro-ecological | Source |
|---|---|---|---|---|
| 1 | TDb/3069 | Togo | Savannah | IITA |
| 2 | TDb/3063 | Togo | Savannah | IITA |
| 3 | TDb/3694 | Congo | Savannah | IITA |
| 4 | TDb/3059 | Togo | Savannah | IITA |
| 5 | TDb/3073 | Togo | Savannah | IITA |
| 6 | TDb/4119 | Guinea | Savannah | IITA |
| 7 | TDb/3119 | Togo | Savannah | IITA |
| 8 | TDb/3082 | Nigeria | Forest | IITA |
| 9 | TDb/3091 | Gabon | Forest | IITA |
| 10 | TDb /3833 | Burkina Faso | Savannah | IITA |
| 11 | TDb/3058 | Togo | Savannah | IITA |
| 12 | TDb/3688 | Nigeria | Forest | IITA |
| 13 | TDb/3080 | Ghana | Forest | IITA |
| 14 | TDb/3065 | Togo | Savannah | IITA |
| 15 | TDb/3096 | Togo | Savannah | IITA |
| 16 | TDb/3693 | Congo | Forest | IITA |
| 17 | TDb/3081 | Gabon | Forest | IITA |
| 18 | TDb/3070 | Togo | Savannah | IITA |
| 19 | TDb/4123 | Sierra Leone | Savannah | IITA |
| 20 | TDb/3060 | Togo | Savannah | IITA |
| 21 | TDb/3834 | Nigeria | Forest | IITA |
| 22 | TDb/1455 | Gabon | Forest | IITA |
| 23 | TDb/3690 | Nigeria | Forest | IITA |
| 24 | TDb/3048 | Benin | Savannah | IITA |
| 25 | TDb/3068 | Togo | Savannah | IITA |
| 26 | TDb/3835 | Nigeria | Swampy with tall tree and grasses | IITA |
| 27 | TDb/3431 | Nigeria | Forest | IITA |
| 28 | TDb/3049 | Benin | Savannah | IITA |
| 29 | TDb/3697 | Congo | Forest | IITA |
| 30 | TDb/3769 | Nigeria | Forest | IITA |
| 31 | TDb/3691 | Congo | Savannah woodland | IITA |
| 32 | TDb/3072 | Nigeria | Forest | IITA |
| 33 | TDb/3071 | Togo | Savannah | IITA |
| 34 | TDb/3045 | Nigeria | Forest | IITA |
| 35 | TDb/3047 | Benin | Savannah | IITA |
| 36 | TDb/3087 | Gabon | Forest | IITA |
| 37 | TDb/2857 | Equatorial Guinea | Forest | IITA |
| 38 | TDb/3078 | Nigeria | Forest | IITA |
| 39 | TDb/3083 | Gabon | Savannah woodland | IITA |
| 40 | TDb/3695 | Congo | Savanah woodland | IITA |
| 41 | TDb/3066 | Togo | Savannah | IITA |
| 42 | TDb/3076 | Togo | Savannah | IITA |
| 43 | TDb/4121 | Sierra Leone | Savannah | IITA |
| 44 | TDb/3061 | Togo | Savannah | IITA |
| 45 | TDb/3085 | Nigeria | Forest | IITA |
| 46 | TDb/3088 | Gabon | Forest | IITA |
| 47 | TDb/3062 | Gabon | Forest | IITA |

*(Continued)*

**Table 1.** (Continued)

| SN | Accession | Origin | Agro-ecological | Source |
|---|---|---|---|---|
| 48 | TDb/4120 | Sierra Leone | Savannah | IITA |
| 49 | TDb/3044 | Nigeria | Forest | IITA |
| 50 | TDb/3090 | Gabon | Savannah woodland | IITA |
| 51 | TDb/3092 | Gabon | Savannah woodland | IITA |
| 52 | TDb/4641 | Togo | Savannah | IITA |
| 53 | TDb/3086 | Gabon | Forest | IITA |
| 54 | TDb/3773 | Nigeria | Forest | IITA |
| 55 | TDb/3079 | Ghana | Forest | IITA |
| 56 | TDb/3190 | Nigeria | Forest | IITA |
| 57 | TDb/3689 | Nigeria | Forest | IITA |
| 58 | TDb /3091 | Gabon | Forest | IITA |
| 59 | TDb/3692 | Congo | Forest | IITA |
| 60 | TDb/3046 | Nigeria | Forest | IITA |
| 61 | TDb/3832 | Burkina Faso | Savannah | IITA |
| 62 | TDb/3067 | Togo | Savannah | IITA |
| 63 | TDb/3075 | Togo | Savannah | IITA |
| 64 | TDb/4122 | Sierra Leone | Savannah | IITA |
| 65 | YA1 | Nigeria | Forest | NRCRI |
| 66 | YA2 | Nigeria | Forest | NRCRI |
| 67 | YA3 | Nigeria | Forest | NRCRI |
| 68 | YA4 | Nigeria | Forest | NRCRI |
| 69 | YA5 | Nigeria | Forest | NRCRI |
| 70 | YB1 | Nigeria | Forest | NRCRI |
| 71 | YB2 | Nigeria | Forest | NRCRI |
| 72 | YB3 | Nigeria | Forest | NRCRI |
| 73 | YB4 | Nigeria | Forest | NRCRI |
| 74 | YB5 | Nigeria | Forest | NRCRI |
| 75 | YC1 | Nigeria | Forest | NRCRI |
| 76 | YC2 | Nigeria | Forest | NRCRI |
| 77 | YC3 | Nigeria | Forest | NRCRI |
| 78 | YC4 | Nigeria | Forest | NRCRI |
| 79 | YC5 | Nigeria | Forest | NRCRI |
| 80 | YD1 | Nigeria | Forest | NRCRI |
| 81 | YD2 | Nigeria | Forest | NRCRI |
| 82 | YD3 | Nigeria | Forest | NRCRI |
| 83 | YD4 | Nigeria | Forest | NRCRI |
| 84 | YD5 | Nigeria | Forest | NRCRI |
| 85 | YE1 | Nigeria | Forest | NRCRI |
| 86 | YE2 | Nigeria | Forest | NRCRI |
| 87 | YE3 | Nigeria | Forest | NRCRI |
| 88 | YE4 | Nigeria | Forest | NRCRI |
| 89 | YE5 | Nigeria | Forest | NRCRI |
| 90 | YF1 | Nigeria | Forest | NRCRI |
| 91 | YF2 | Nigeria | Forest | NRCRI |
| 92 | YF3 | Nigeria | Forest | NRCRI |
| 93 | YF4 | Nigeria | Forest | NRCRI |
| 94 | YF5 | Nigeria | Forest | NRCRI |

### Field establishment

The experimental design used was the randomized complete block design (RCBD). The planting was arranged in ridges with planting at the beginning of the rainy season. A single row plot was used for each accession, each row measuring 7m long with 1m spacing between rows and 1m spaces between plants within a row. Each accession was planted in two replication for one cropping season. This spacing regimen was done to avoid competition among neighbouring plants and to ensure sound establishment of each accession. Individual plants were supported by bamboo stakes. Standard agronomic practices such as manual weeding, with no irrigation, no chemical was adopted. Phenotypic data was collected on the five middle plants from each row. A total of 10 quantitatives and 5 qualitatives traits was used to profile the 94 accessions.

### Phenotyping

The trais described in Table 2 were used to assess the agronomic performance of the 94 yam accessions. Data was recorded using the yam standard operational protocol developed by [36] with slight modification.

### Genotyping

**Yam leaf sampling and DNA extraction.** Young, healthy and fully expanded fresh leaf samples were collected at two months after the all accessions were fully established. About three to five tender leaves, weighing more than 20mg were collected in well labelled bags containing silica-gel granules with a colour indicator. The leaf samples were stored in the silica-gel for 72 hours to remove the moisture. Subsequently, genomic DNA (gDNA) extraction was carried out at the Bioscience centre, International Institute of Tropical Agriculture (IITA), Ibadan, Nigeria, using the CTAB procedure with slight modification [37]. The DNA quality and concentration was ascertained by running the gDNA in a 1% agarose gel and on a NanoDrop 2000 spectrophotometer, following the methods described in [38].

**SNP genotyping assay and SNP filtering.** High quality DNA was sent to Diversity Array Technology (DArT) Australia for the sequencing. For the sequencing-based DArT genotyping,

**Table 2. Morphological descriptors used for phenotypic characterization of *D.bulbifera* yam accessions.**

| Variables | traits | Codes | Descriptors |
|---|---|---|---|
| Plant height (m) | Quntitative | PH | Mean height of the selected plant at four weeks from sowing |
| Number of stem per plant (No) | | SPP | Mean number of stem per plant of the selected plants |
| Number of branches on main stem (No) | | NBMS | Mean number of branches on the main stem per plant of the selected plants |
| Leaf length (cm) | | LLENGTH | Mean length of the selected leaf in f the selected plants |
| Leaf width (cm) | | LWIDTH | Mean width of the selected leaf in the selected plants |
| Bulbil diameter (mm) | | BDIA | Average diameter of one bulbil from the selected plants |
| Bulbil width (cm) | | BWIDTH | Average width of one bulbil from the selected plants |
| Bulbil Length(cm) | | BLength | Average weight of one bulbil from the selected plants |
| Total number of bubils | | TNB | |
| Non marketable | | Non.mkt | |
| Leaf margin colour | Qualitative | LMCOL | 1-Green; 2-Purple; 3-Other |
| Leaf hairiness | | LH | 1-Upper surface; 2-Lower surface; 3-Both |
| Leaf shape | | LS | 1-Ovate;2-Cordate long;3-Cordate broad;4-Sagittate long;5-Sagittate broad;6-Hastate; 99-Other |
| Leaf apex shape | | LAS | 1-Optuse; 2-Acute; 3-Emarginate; 99-Other |
| Bulbils shape | | BS | 1-Round; 2-Oval; 3-Irregular; 4-Elongate |
| Yam anthracnose disease | | YAD | 1-Highly resistant; 2-resistant; 3-Moderately resistant; 4-Susceptibe; 5-Highly susceptible |

SNPs were called using complexity reduction methods optimized for yam at the DArT's proprietary software, DArTSoft, as described by [39]. We aligned the raw reads to the yam reference genome [40].

A total of 11,721 DArTseq SNP-derived markers were obtained as raw SNP markers and subjected to quality control (QC) to eliminate the unwanted SNP markers. For the QC implementation, software PLINK [41] version 1.9 and VCFtools [42] were used. Markers and genotypes with high missing value (>20%) were eliminated as well as SNP marker with low minor allele frequency (<5%) and low call rate. After quality filtering, 10,087 DArT-SNP markers distributed across 20 yams chromosomes were retained and used for downstream analysis. Genotypic data can be downloaded from the downloaded from the open access YamBase (www.yambase.org).

**Phenotypic data analysis.** Quantitative data collected were subjected to descriptive statistical analysis (minimum, maximum, average and standard error) using basic function in R. Principal component analysis (PCA) was performed using FactorMiner package [43]. The PCA data was used to generate eigenvalues, cumulative variability, and load coefficient values. The principal components (PC) with eigenvalues > 1.0 were selected, and those traits that had load coefficients ≥ 0.5 were considered relevant scores for the PC and considered as valuable traits for distinguishing between the genotypes [44].

**Assessment of genetic diversity.** Using VCFtools [42] and PLINK 1.9 [41] minor allele frequency (MAF), polymorphic information content (PIC), expected heterozygosity (He), and observed heterozygosity (Ho) were estimated. For Genetic diversity analysis we first estimated the optimal number of clusters by using k-means analysis [45, 46] of PCA (principal component analysis)-transformed genome-wide SNP data by varying the possible number of clusters from 1 to 10. Population structure analysis was conducted through ADMIXTURE analysis using adegenet R package. Ancestry probability was used to determine the most appropriate K, and accessions with membership proportions (Q-value) ≥ 60% were assigned to groups, while those with membership probabilitie s less than 60% were considered as admixt [47]. To complement the the PCA and population structure, hierachical cluster (HC) analysis was conducted using Jaccard dissimilarity matrix. Clusters generated for both genotypic and phenotypic were compared using dendextend package.

## Results

### Variability based phenotypic data

The result of the phenotypic statistics, including mean, minimum, range, standard deviation and maximum are shown in Table 3. High phenotypic variation was observed for quantitative traits. The accessions exhibited significant variability for most of the traits assessed. Trait such bulbil width displayed wide range from 3 to 8 with 5 as average while the number of branches on main stem ranged from 3 to 17 with 9.50 as average value. Similarly. For the yam leaf length, it ranged from 10 to 16 cm among the accessions with a mean value of 13.59cm.

Through the PCA, the first three components explained 73.31% of the total variation (Table 4). The first principal component (PC1) had an Eigen value of 9.17, contributing to 57.11% of the total variation. On the first component, all evaluated variables displayed high variation except yam leaf margin color and the yam anthracnose disease. On the second principal component which explained 8.80 of the total vaiation the leaf margin color and the anthracnose disease were identified to be highly associated.

The influence of the traits on the principal components and the levels of correlation among them are presented in Fig 1. Traits like Leaf shape (LS), Bulbils shape (BS), Plant height (PH), Number of stem per plant (SPP), Number of branches on main stem (NBMS), Llengh. Lwidth,

**Table 3. Descriptive statistics of some quantitative morphological traits of *D. bulbifera*.**

| Variables | Mean | Sd | Min | Max | Range | Skew | Kurtosis |
|---|---|---|---|---|---|---|---|
| LMCOL | 1.265 | 0.300 | 1.000 | 2.000 | 1.000 | 0.927 | -0.029 |
| LH | 2.386 | 0.678 | 1.000 | 3.000 | 2.000 | -0.597 | -1.073 |
| LS | 2.354 | 0.364 | 2.000 | 3.000 | 1.000 | 0.538 | -1.103 |
| LAS | 2.508 | 0.497 | 1.667 | 3.000 | 1.333 | -0.089 | -1.925 |
| BS | 2.918 | 0.670 | 1.000 | 4.000 | 3.000 | -0.633 | -0.300 |
| PH | 1.955 | 0.487 | 1.180 | 2.887 | 1.707 | 0.283 | -1.328 |
| SPP | 6.220 | 2.647 | 1.333 | 10.000 | 8.667 | 0.203 | -1.577 |
| NBMS | 9.500 | 3.346 | 4.667 | 17.667 | 13.000 | 0.418 | -0.997 |
| LLENGH | 13.590 | 1.308 | 10.810 | 16.463 | 5.653 | -0.218 | -0.637 |
| LWIDTH | 10.716 | 1.519 | 7.457 | 13.700 | 6.243 | -0.289 | -0.888 |
| Bdia | 34.298 | 9.216 | 20.170 | 55.500 | 35.330 | 0.230 | -1.001 |
| non.mkt | 34.016 | 20.029 | 8.667 | 98.667 | 90.000 | 1.194 | 0.683 |
| TNB | 41.344 | 12.495 | 19.667 | 80.000 | 60.333 | 0.675 | 0.139 |
| Blenght | 8.839 | 1.711 | 5.390 | 11.627 | 6.237 | -0.464 | -1.149 |
| Bwidth | 5.385 | 1.447 | 3.167 | 8.713 | 5.547 | 0.230 | -1.001 |
| YAD | 1.206 | 0.257 | 1.000 | 2.000 | 1.000 | 0.956 | -0.007 |

Sd: Standard deviation; Min: Minimum; Max: Maximum

Relative contribution of individual traits to phenotypic variation.

non marketable (Non.MKT), leaf length (LLENGH), leaf width (LWIDTH) and Total number of bubils (TNB) were positively associated while traits like LH, LAS, BDia, Blenght, Bwidth) were all negatively associated.

Correlation analysis revealed positive correlation among several of the evaluated traits, the vine length displayed positive corelation with the stem per plant and with the number of the branches per stem. Also, and through the correlation analysis positive colleration was reported between the leaf length, the number of branches per plant, the stem number per plant and the vine length while negative correlation was observed between the stem diameter and the number of stem per plant (Fig 2).

## Phenotypic classification

Using 16 traits, the evaluated yam accessions were grouped into 2 major groups (Fig 3). The first cluster (green) was made of only accessions collected from IITA genebank. Accessions of this cluster are characterized with low yield. The second cluster (red) was made with accessions collected from different farmers field across the South East of Nigeria. Accession of this cluster were identified to have high number of bulbils with large tubers.

## Summary statictic based SNP markers

A total of 11,721 SNP markers was initially generated, from which 10,087 was retained after the removal of low-quality SNP markers. The SNP markers were unequally distributed across the twenty (20) chromosomes of *D. bulbifera* (Table 5). The highest number of SNPs (516) was mapped on chromosome 10 and the lowest number of SNPs (489) was mapped on chromosome 15 (Table 5). Transition SNPs (62.39%, 6293 SNPs) were more frequent than transversions SNPs (37.61%, 3794 SNPs). The C/T transitions (31.14%) accounted for the highest frequency, while C/G transversions (6.77%) occurred at the lowest frequency (Fig 4). The average polymorphic information content (PIC) value across all the markers was 0.268, it ranged

**Table 4.  The proportion of the morphological variation and traits contribution explained by the first three (3) principal components.**

| Variables | PC1 | PC2 | PC3 | PC4 | PC5 |
|---|---|---|---|---|---|
| LMCOL | -0.05 | 0.70 | -0.01 | 0.50 | 0.44 |
| LH | -0.85 | 0.05 | -0.02 | 0.20 | -0.25 |
| LS | 0.72 | -0.08 | -0.14 | -0.22 | 0.17 |
| LAS | -0.94 | 0.01 | 0.09 | 0.12 | -0.14 |
| BS | 0.74 | 0.12 | -0.09 | -0.02 | 0.22 |
| PH | 0.89 | -0.02 | 0.02 | -0.10 | -0.03 |
| SPP | 0.91 | 0.13 | 0.14 | -0.17 | 0.13 |
| NBMS | 0.86 | 0.12 | 0.05 | -0.21 | 0.20 |
| LLENGH | 0.68 | -0.34 | 0.46 | 0.32 | 0.08 |
| LWIDTH | 0.65 | -0.38 | 0.34 | 0.48 | -0.01 |
| Bdia | -0.85 | 0.04 | 0.43 | -0.16 | 0.18 |
| non.mkt | 0.73 | 0.18 | 0.37 | -0.01 | -0.25 |
| TNB | 0.68 | 0.24 | 0.40 | -0.18 | -0.29 |
| Blenght | -0.81 | 0.18 | 0.36 | -0.26 | 0.16 |
| Bwidth | -0.85 | 0.04 | 0.43 | -0.16 | 0.18 |
| YAD | -0.36 | -0.69 | 0.00 | -0.08 | 0.37 |
| eigenvalue | 9.14 | 1.41 | 1.18 | 0.93 | 0.79 |
| variance | 57.11 | 8.80 | 7.40 | 5.80 | 4.96 |
| cumulative | 57.11 | 65.91 | 73.31 | 79.12 | 84.07 |

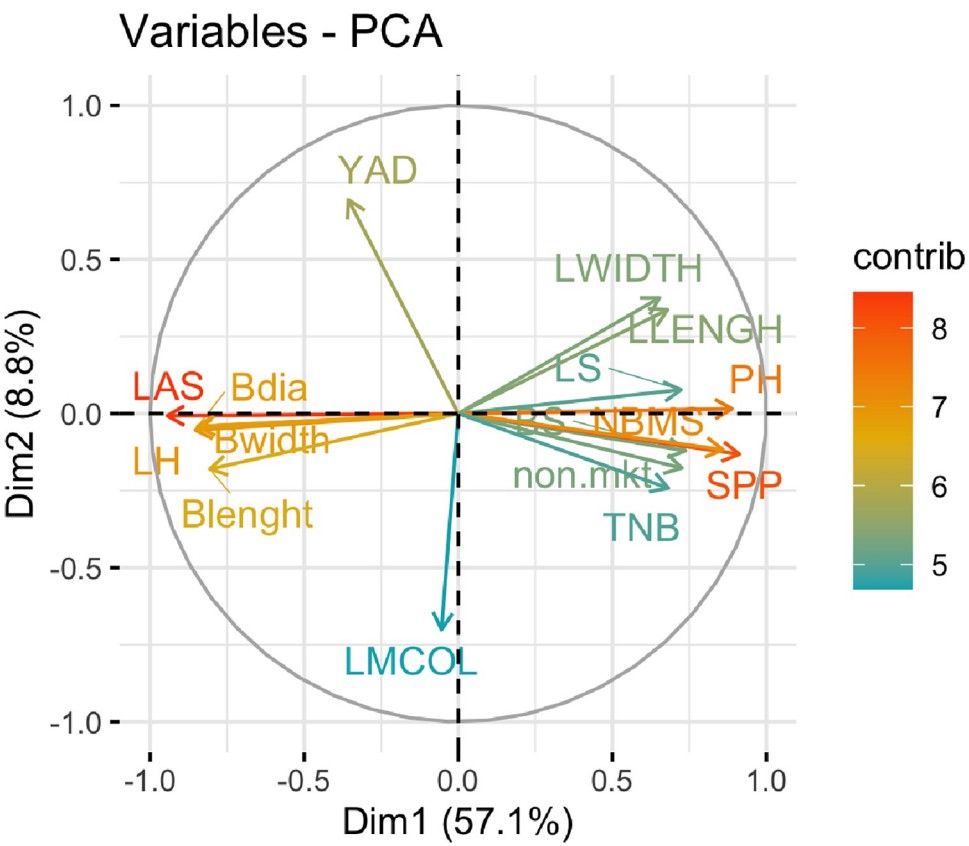

**Fig 1.  Principal component analysis plot showing the total contribution of variables accounting for variability in PC1 and PC2.**

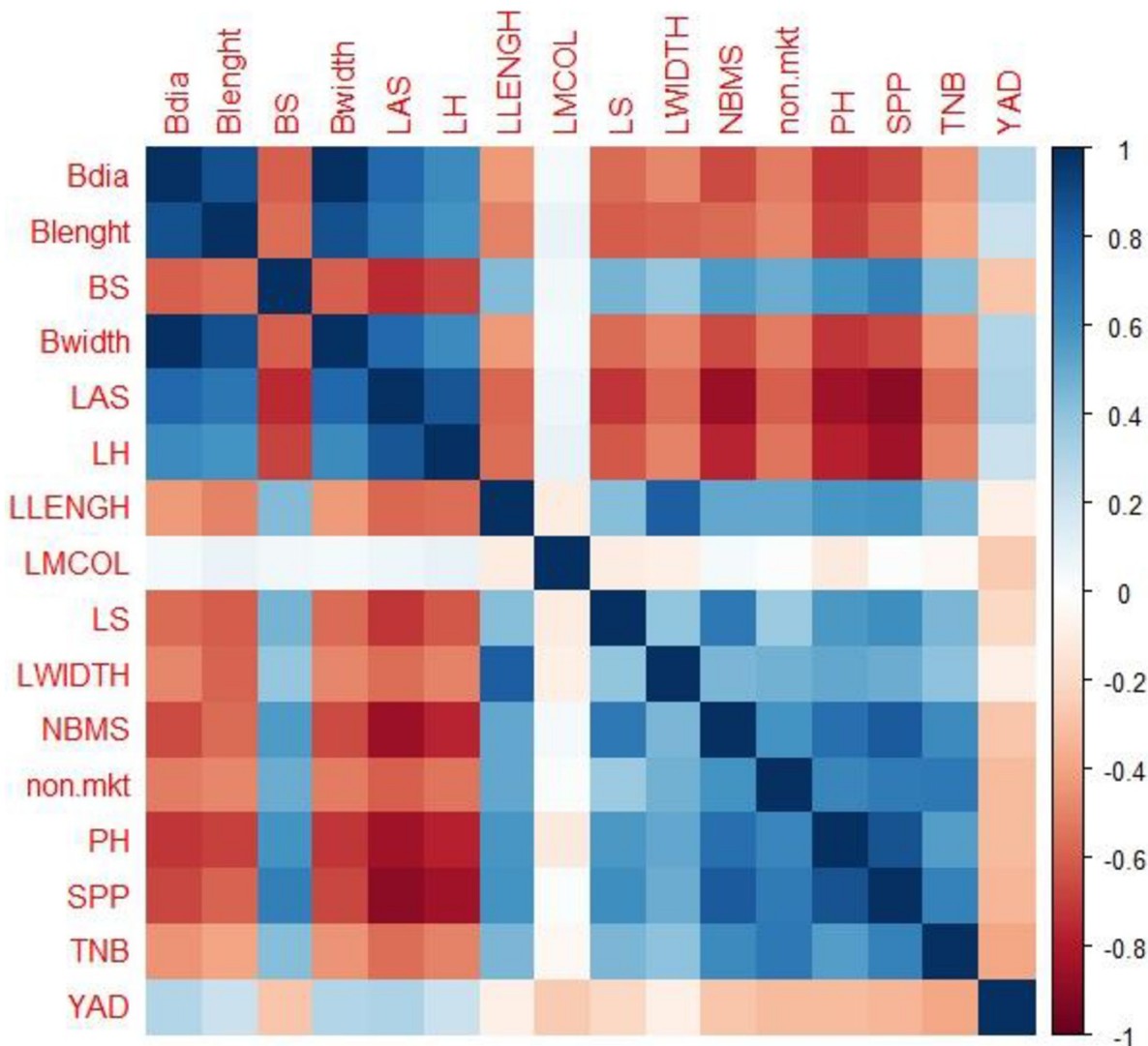

**Fig 2. Phenotypic correlation among the evaluated traits.**

from 0.262 to 0.277 while the observed heterozygosity (Ho) ranged from 0.252 to 0.278 with an average of 0.262 (Table 5, Fig 5). The expected heterozygosity (He) ranged between 0.320 and 0.342, and the mean genetic diversity was 0.329. Similarly, the minor allele frequency (MAF) ranged between 0.226 and 0.249 with an average of 0.235 (Fig 5).

## Population structure

Through the Bayesian Information Criterion (BIC) and complementary coordination analysis, three clusters was identified as the optimum number of the possible group number (Fig 6). The 94 accessions were classified into three (3) main groups (Fig 7) through phylogenic analysis. The first cluster (red), which were all made up of IITA genebank accessions. A larger number of accessions were in the second cluster (green) containing forty five (45) accessions (47.87%) of which 40 were IITA genebank accessions and 5 were different accessions obtained from different communities within Anambra State of Nigeria. The third Cluster (blue) was made of twenty six (26) accessions (27.66%) which comprised mainly of the accessions

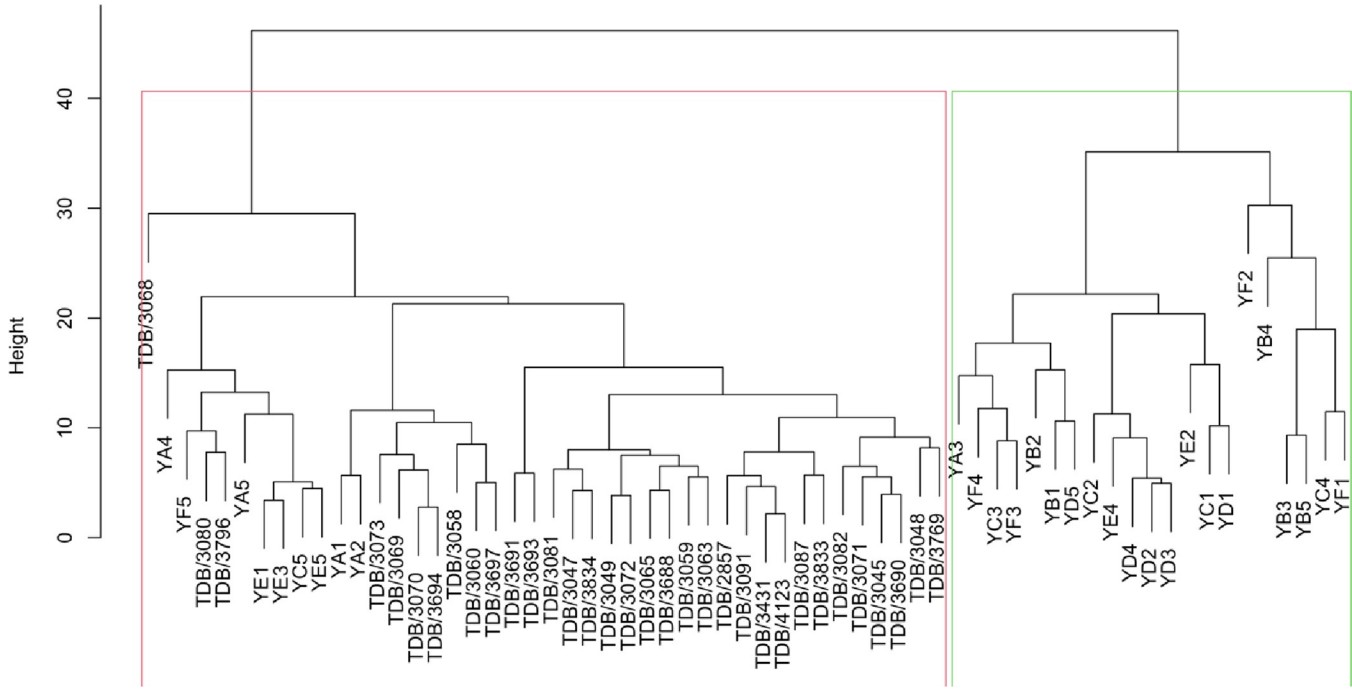

**Fig 3. Hierachical clustering based phenotypic data.**

obtained from other different south eastern states of Nigeria and one (1) IITA genebank accession (Fig 7). Through the admixture analysis, most of the accessions were properly assigned to a genetic group and only a few were considered as admixt (Fig 8). Out of the 94 accessions, only seven accessions (TDB 3045, TDB 3087, TDB 3049, TDB 3835, TDB 3119, TDB 3695) displayed ancestry probability < 50% and were considered as admixt. Simulations (logarithm probability relative to standard deviation, ΔK) estimated from the SNP markers showed a sharp peak at K = 3 which explained the optimum number of sub-populations (ΔK = 3). At ΔK = 3, cluster I, cluster II and cluster III consisted of 26 accessions (27.66%), 45 accessions (47.87%), and 23 accessions (24.47%) respectively. Molecular variance analysis (AMOVA) revealed low genetic diversity within genetic group with the highest genetic diversity been observed among the genetic group.

## Clustering based phenotypic and genotypic data

Clustering based on phenotypic data revealed the presence of two major clusters, whereas molecular marker analysis grouped the evaluated yam accessions into three distinct clusters. When comparing the clustering methods, none of the yam accessions were positioned in the same cluster, given the 0.29 entanglement threshold (Fig 9) comparision of genotypic and phenotypic.

## Discussion

In this study, we evaluated the level of the genetic diversity among 94 accessions of *D. bulbifera* using both phenotypic and molecular markers. A better understanding of the existing aerial yam germplasm is one of the prerequisites for breeding new genotypes with novel or improved

**Table 5. Summary statistics of SNP markers across 20 chromosomes of *D. bulbifera* accessions.**

| Chr | SNP no | Ho | He | PIC | MAF |
|---|---|---|---|---|---|
| 1 | 511 | 0.268 | 0.342 | 0.277 | 0.249 |
| 2 | 509 | 0.262 | 0.330 | 0.269 | 0.235 |
| 3 | 490 | 0.257 | 0.326 | 0.266 | 0.232 |
| 4 | 509 | 0.264 | 0.325 | 0.265 | 0.233 |
| 5 | 511 | 0.257 | 0.323 | 0.264 | 0.227 |
| 6 | 507 | 0.268 | 0.328 | 0.267 | 0.233 |
| 7 | 507 | 0.271 | 0.337 | 0.273 | 0.243 |
| 8 | 506 | 0.278 | 0.330 | 0.268 | 0.237 |
| 9 | 507 | 0.256 | 0.320 | 0.262 | 0.226 |
| 10 | 516 | 0.257 | 0.322 | 0.263 | 0.229 |
| 11 | 514 | 0.271 | 0.328 | 0.267 | 0.233 |
| 12 | 494 | 0.262 | 0.325 | 0.265 | 0.231 |
| 13 | 492 | 0.256 | 0.330 | 0.269 | 0.233 |
| 14 | 513 | 0.256 | 0.329 | 0.268 | 0.235 |
| 15 | 489 | 0.263 | 0.340 | 0.276 | 0.242 |
| 16 | 505 | 0.267 | 0.332 | 0.270 | 0.237 |
| 17 | 510 | 0.257 | 0.334 | 0.271 | 0.239 |
| 18 | 510 | 0.255 | 0.324 | 0.264 | 0.228 |
| 19 | 498 | 0.252 | 0.326 | 0.266 | 0.232 |
| 20 | 509 | 0.267 | 0.333 | 0.270 | 0.242 |
| Total/Average | 10,087 | 0.262 | 0.329 | 0.268 | 0.235 |

SNP: Single nucleotides polymorphism; HO: Observed heterozygosity; He: Expected heterozygosity; PIC: Polymorphic information content; MAF: Minor allele frequency.

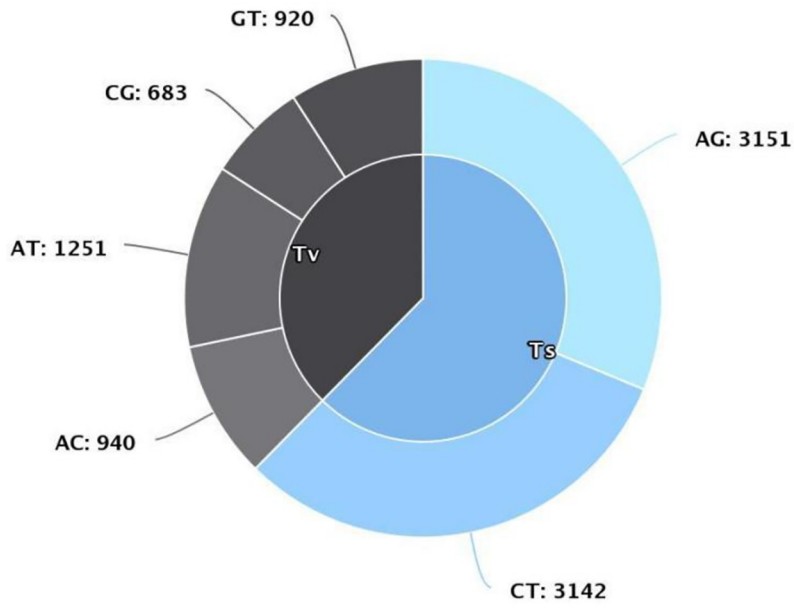

**Fig 4. Transition and transversion based on bi-allelic SNP markers.** Tv: Transversions; Ts: Transitions; A: Adenine; T: Thymine; G: Guanine; C: Cytosine. Chart developed using SNIPLAY software.

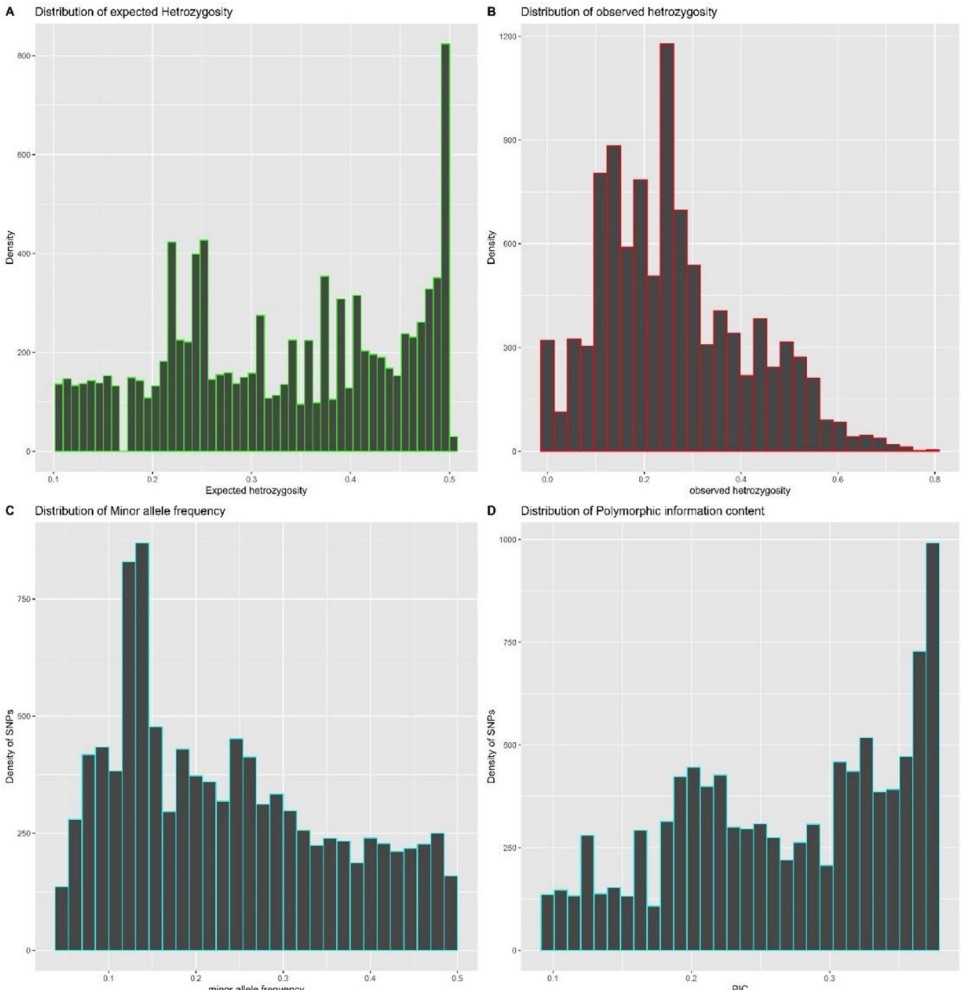

**Fig 5. Histogram showing the distribution of SNPs markers associated with D.bulbifera genome.** A) Distribution of expected heterozygosity in the genome B) Distribution of observed heterozygosity in the genome. C) Distribution of Minor allele frequency in the genome. D) Distribution of polymorphic information content.

characteristics. The principal component derived from the study of these traits displayed 73.17% of the total phenotypic on the first three components coupled with large differences between minimum and maximum confers the strength of these traits for traits profiling. Observed variability could be as a result of sexual recombination [48], and natural mutation and long term selection occurring in the course of its ongoing domestication process in some agro-ecological zones [49–52]. Despite the potential of phenotypic traits in diversity studies, their expression may be partly subjected to environmental variation, thus, providing limited genetic information [18].

The DArTseq genotyping detected a total of 10,087 informative SNPs, which were unequally distributed among and within the 20 aerial yam chromosomes at various densities. The average PIC value of 0.27 obtained in the present study is higher than previous reports [53] but comparable to some other studies [54–56]. This shows the informativeness of the SNP markers used in this study.

Through structure and phylogenetic tree analyses, the 94 aerial yam accessions used in this study were classified into three subgroups. For all the subgroups, both accessions obtained

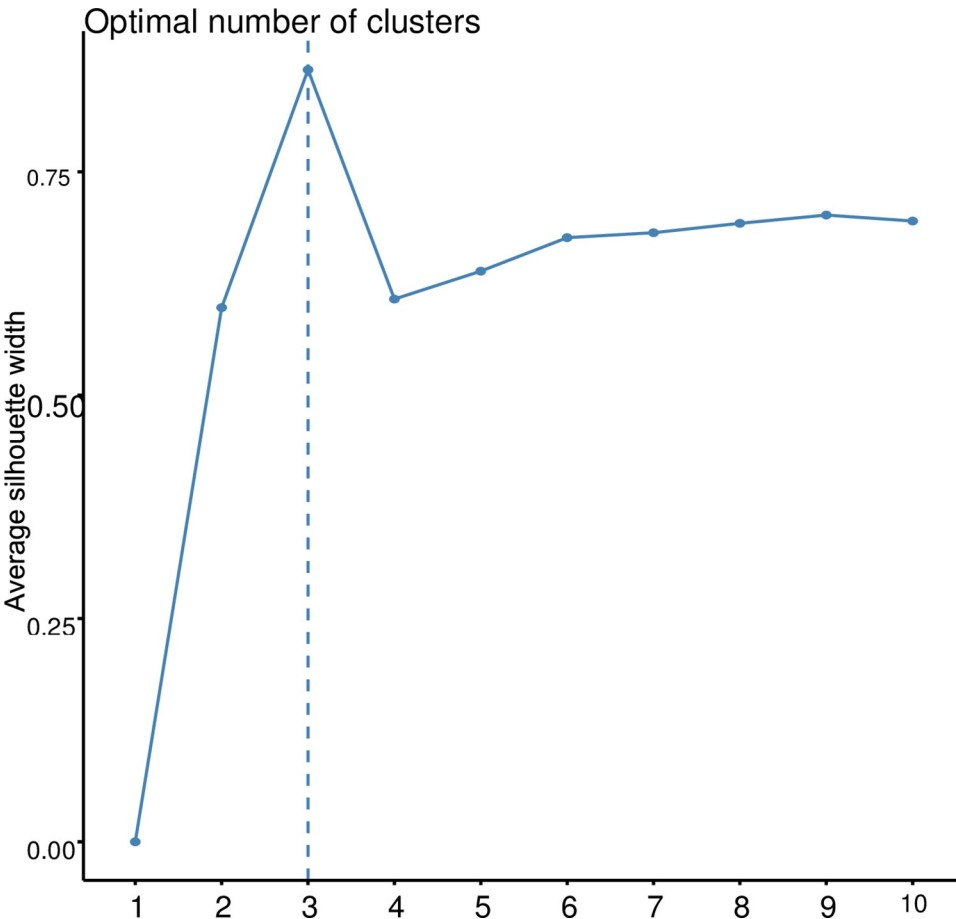

**Fig 6. Graph representing the estimated membership fraction using LnP-(D) derived delta K with clusters number (K) ranged from 1–10 for K = 3.**

from IITA genebank and the eastern part of Nigeria were well distributed and showed high correspondence in clustering patterns between the different grouping methods. Similar results were observed in *Dioscorea alata* [35]. The three subgroups of aerial yam accessions was observed to exhibit a low level of admixture. These observed genetic divergence with a low admixture level is due to original differences in domestication and subsequent vegetative propagation by farmers. Yam clones in farmer's field are genetically homogenous with negligible recombination rates, as farmers select their planting material from tubers and not from botanical seed. However, the molecular evidence on ennobled cultivars showed that the tubers collected by farmers from wild environments are often a mixture of wild and interspecific hybrids [40, 57, 58]. This could partly explain the origin of the genetic admixture identified among the aerial yam accessions. Harnessing the advantages of phenotypic and molecular markers improves the grouping of entries in a germplasm collection [59], which provides a piece of valuable base information for parental selection to realize and sustain genetic gain. To our knowledge, our study represents the first high throughput genotyping of *D. bulbifera* and thus highlights the valuable prospect of utilizing DArTseq-SNP markers for molecular genetic studies in yams. Our findings buttress the fact that there are in and across country related variations. The studied genotypes were collected from different countries and diverse locations in Nigeria. The selected SNP markers resulted to three cluster groups. Two independent cluster

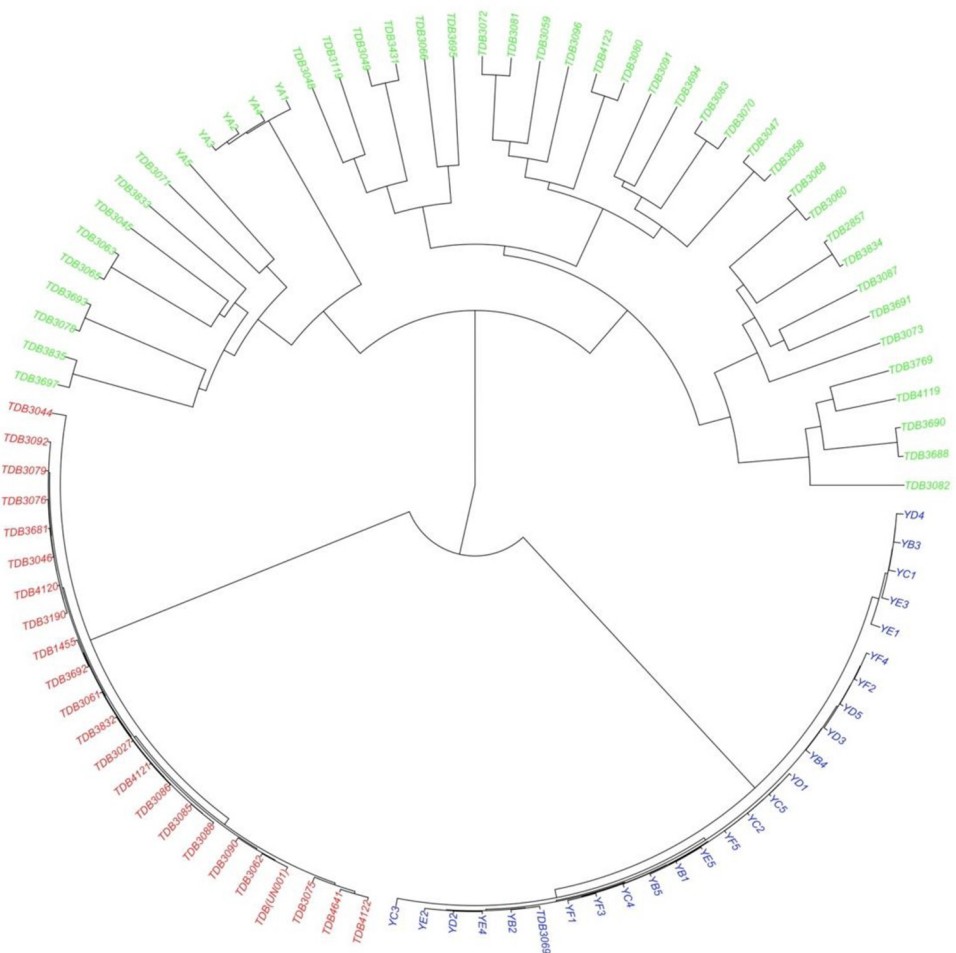

**Fig 7. Hierarchical circular clustering dendrogram generated using the UPGMA method and Jaccard's dissimilarity matrix.** Different colours indicate different groups identified: Cluster 1 (red), Cluster 2 (green), Cluster 3 (blue).

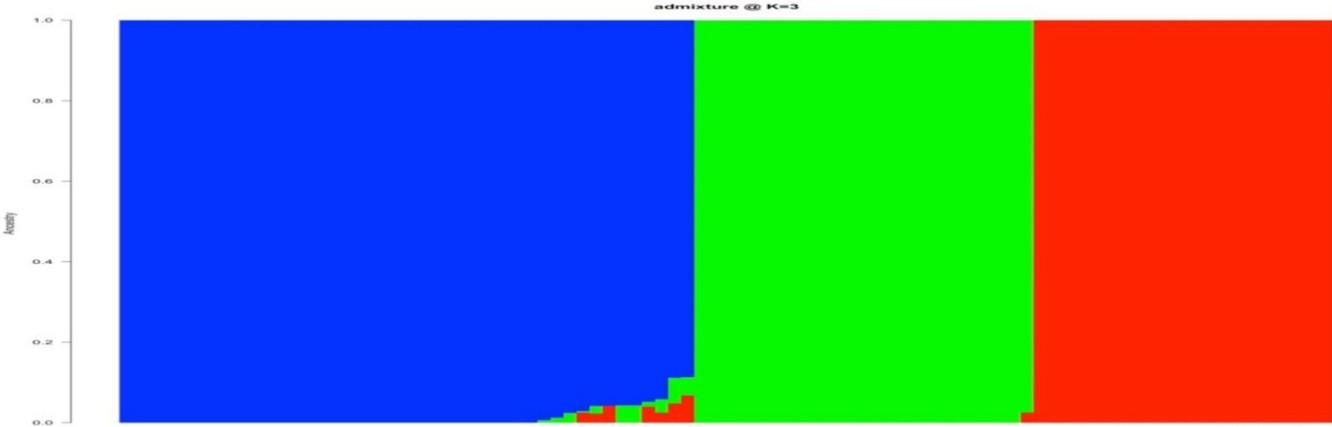

**Fig 8. Grouping pattern in 94 accessions of *D. bulbifera* at K = 3 based on the Bayesian clustering method.** The colour displays each cluster: Blue (cluster 1), Green (cluster 2) and Red (cluster 3). Each vertical bar corresponds to an accession and the colour proportion in each bar represents the probability of each accession being affiliated to the different clusters.

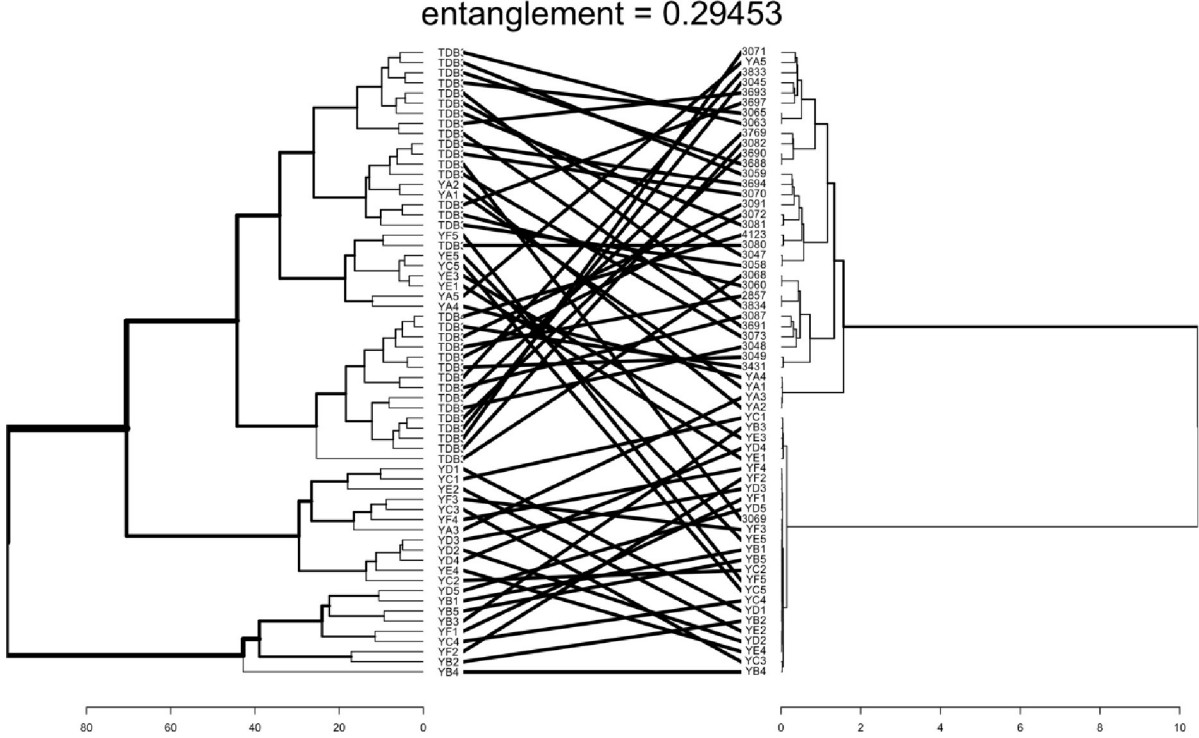

**Fig 9. Clustering comparision of genotypic and phenotypic.**

groups comprised mainly individuals from IITA genebank while a third cluster comprised mostly of cultivars obtained from South Eastern states of Nigeria. The trend in variability infers evolutionary and admixture relationship as reported by Adjei [31]. Cluster II had individuals from different countries suggesting some sort of clonal dispersion though trade and migration within the African sub region. We observed negligible association between the phenotypic and genotypic data which could be a result of high variation in phenotypic traits. In a similar study, Agre et al. [32, 34] reported similar result on *D. alata* and *D. rotundata* in a study conducted in Nigeria and Benin. The low correlation between morphological and molecular data in the *D. bulbufera* accessions generally suggests that the two data types are appropriate for a combined use, which can deepen understanding and discriminate the genotypes better due to the non-overlapping information.

This implies the existence of a robust genepool from which breeding programs can thrive to make selection and breeding advance. It is noteworthy to highlight that breeding programs on yam has prioritised *D. rotundata* and *D.alata* in the past. This conscious effort sets the very foundation for *D. bulbifera* crop improvement strategy.

## Conclusion

In this study, we investigated the genetic diversity of *Dioscorea bulbifera* using agronomic phenotypic and DArTseq SNP markers. This study revealed the presence of moderate genetic diversity among the evaluated genotypes. Genetic diversity through the population structure and hierarchical clustering analysis displayed same genetic pattern in the grouping. We provided as well the relevance of combining phenotypic data with molecular markers for proper genetic profiling.

## Acknowledgments

We are grateful for the technical support the yam breeding team of NRCRI provided in field data collection. We acknowledge the yam team of the IITA Bioscience center for their laboratory support.

## Author Contributions

**Conceptualization:** Eunice Ekaette, Paterne A. Agre.

**Data curation:** Eunice Ekaette, Ikenna Nnnabue, Paterne A. Agre.

**Formal analysis:** Eunice Ekaette, Ikenna Nnabue, Paterne A. Agre.

**Funding acquisition:** Jude Obidiegwu, Paterne A. Agre.

**Methodology:** Jude Obidiegwu, Paterne A. Agre.

**Resources:** Paterne A. Agre.

**Software:** Paterne A. Agre.

**Supervision:** Peter Okocha, Paterne A. Agre.

**Validation:** Paterne A. Agre.

**Visualization:** Paterne A. Agre.

**Writing – original draft:** Eunice Ekaette, Jude Obidiegwu, Paterne A. Agre.

**Writing – review & editing:** Eunice Ekaette, Emeka Nwofia, Peter Okocha, Ikenna Nnnabue, Kenneth Eluwa, Jude Obidiegwu, Paterne A. Agre.

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
