## [Decision Letter · Decision Letter 0]

5 Nov 2023

PONE-D-23-28562Exploring the Genetic Diversity and Population Structure of Aerial yams (Dioscorea bulbifera L.) DArT-seq and agronomic traits.PLOS ONE

Dear Dr. Agre,

Thank you for submitting your manuscript to PLOS ONE. After careful consideration, we feel that it has merit but does not fully meet PLOS ONE’s publication criteria as it currently stands. Therefore, we invite you to submit a revised version of the manuscript that addresses the points raised during the review process.

We look forward to receiving your revised manuscript.

Kind regards,

Timothy Omara, PhD

Academic Editor

PLOS ONE

Journal Requirements:

https://pureadmin.qub.ac.uk/ws/files/470638768/world-04-00020.pdf

https://www.mdpi.com/2037-0164/14/1/25/html

https://horizon.documentation.ird.fr/exl-doc/pleins_textes/divers18-12/010074472.pdf

In your revision ensure you cite all your sources (including your own works), and quote or rephrase any duplicated text outside the methods section. Further consideration is dependent on these concerns being addressed.

Reviewers' comments:

Reviewer's Responses to Questions

**Comments to the Author**

1. Is the manuscript technically sound, and do the data support the conclusions?

Reviewer #1: Yes

Reviewer #2: Yes

2. Has the statistical analysis been performed appropriately and rigorously? 

Reviewer #1: Yes

Reviewer #2: Yes

3. Have the authors made all data underlying the findings in their manuscript fully available?

Reviewer #1: Yes

Reviewer #2: Yes

4. Is the manuscript presented in an intelligible fashion and written in standard English?

Reviewer #1: Yes

Reviewer #2: Yes

5. Review Comments to the Author

Reviewer #1: Introduction

Language structure issues in “…popular staples in West Africa [3] while serving as important source of dietary calories thus contributing on average, more than 200 Kilocalories per person per day to ….“ the sentence could be better written as “….popular staples in West Africa [3], serving an important source of dietary calories with contribution of 200 Kilocalories on average per person per day to o…”

Line 5 says “… diverse agro-ecologies of the country …”, What country? Nigeria? If West Africa please refer to it as you have not mentioned any country at this point.

Methods

The authors have not stated how long (days) the accessions were planted before harvesting

The authors made mention of agronomic practices regularly applied to the crop, however, agronomic practices are too vague and the author should state these agronomic practices in supplementary if it will disrupt the section structure

For the DNA extraction, the authors stated they use young leaves, how are the young leaves selected/identified? (i.e. was it the youngest leaves from the yam or the youngest leaves from a collar, trifoliate, etc.). This is vital to grade uniformity of the harvest tissues because if younger leaves were selected randomly in the yam from accessions to accession, then the data's reproducibility is questioned.

Results

In Table 3, what does the a, b, c, d … represent? does it mean how statistically different the traits are? Indicate this in the table legend.

Figure 1 needs better resolution, the labels are not clearly visible

Figure 2 legend needs to be detailed better

The heat map section should be given careful attention, the authors need to rearrange the letter in a clear way and specify the most significant. The heat map figures need better resolution as the accession could hardly be seen. The authors may only plot the map of the top most significant variation and include the others/this as supplementary material.

Discussion

The discussion therein contains a detailed explanation of the results and the authors clearly identified the gap, however, the discussion section needs to be well-arranged with enough paragraphs separating different objectives and interrelating results. As it is, the discussion is vague, and if possible should have subheadings.

Conclusion

The conclusion is too minimal and a more robust conclusion highlighting the significant high impact of this work needs to be established. The last sentence for instance can be improved upon.

Reviewer #2: The manuscript “Exploring the Genetic Diversity and Population Structure of Aerial yams (Dioscorea bulbifera L.) DArT-seq and agronomic traits” presents insightful knowledge on the diversity among the bilbifera species of yam.

To make the manuscript more presentable the following comments can be taken into consideration:

1. Abstract: Very simplified but can be improved

2. Introduction:

a. The introduction needs some more literature that will cut across some findings and the importance of the use of SNPs markers for diversity studies.

b. The production and distribution of the yam specie used for the study should be emphasized with facts.

c. Check the formatting.

d. Is there any study on D. bulbifera? Some highlights can be in the introduction.

e. Other minor typo errors have been indicated in the attachment.

3. Methodology:

a. Kindly provide some information on the nature of the soil for the experimental site.

b. Table 1: Kindly indicate the accessions that were samples from IITA and NRCRI in the table.

c. Kindly indicate the number of replications used for this design.

d. “A total of 7 qualitative morphological descriptors for yam was used to profile the 94 accessions” This statement can begin for the Phenotyping session. From the statement here, you indicated seven but the parameters in the table are 8. Please clarify and make the necessary adjustments. This may affect the analysis, and this must be clear in the results.

e. I suggest tuber parameters should be added to give a very clear diversity in the accessions used.

f. Pictures that show clear morphological variation can also be very good to show the difference observed.

g. Table 2: Indicate the period in which the parameters were taken.

4. Results:

a. Table 3: This table does not reflect the number of parameters in Table 2. Please clarify and make the necessary adjustments.

b. In the abstract author indicated AMOVA results but that is not reflected in the results. I suggest the AMOVA results are indicated and explained.

c. Figure 3. Explanations about this figure were not stated in the results. It is important to update that.

d. Figures 7 and 8 must be together since they communicate the same information.

e. Please note this well: To improve on the results and analysis. Further analysis can be done by combining the phenotypic and genotypic information in one analysis “dendrogram” to show the relationship and to support the information acquired for both data. That will be very good and explanatory enough. As it stands now, the phenotypic and genotypic information are separated and hanging but it will be good to blend the information from both sides together to make the study rich.

5. Discussion:

a. The authors have elaborated on the required literature, however there is no discussion on the implication of the morphological observations with the marker’s information identified or vice versa.

b. I suggest the authors explain more about that after combining the analysis of the results as indicated in the results session.

6. PLOS authors have the option to publish the peer review history of their article (what does this mean?). If published, this will include your full peer review and any attached files.

Reviewer #1: No

Reviewer #2: No

---

## [Author Response · Author response to Decision Letter 0]

23 Apr 2024

Journal Requirements:

Query 

Response 

We appreciate the feedback from the editors we have formatted the manuscript to the PLOS requirement

https://pureadmin.qub.ac.uk/ws/files/470638768/world-04-00020.pdf

https://www.mdpi.com/2037-0164/14/1/25/html

https://horizon.documentation.ird.fr/exl-doc/pleins_textes/divers18-12/010074472.pdf

In your revision ensure you cite all your sources (including your own works), and quote or rephrase any duplicated text outside the methods section. Further consideration is dependent on these concerns being addressed.

Response 

We have adjusted the entire manuscript to avoid any form of overlapping text. We have as well added new references 

Response

Thanks for the notice we have provided the correct grant number in the revised manuscript 

Reviewer's Responses to Questions

Reviewers' comments:

Comments to the Author

1. Is the manuscript technically sound, and do the data support the conclusions?

Reviewer #1: Yes

Reviewer #2: Yes

2. Has the statistical analysis been performed appropriately and rigorously?

Reviewer #1: Yes

Reviewer #2: Yes

3. Have the authors made all data underlying the findings in their manuscript fully available?

Reviewer #1: Yes

Reviewer #2: Yes

4. Is the manuscript presented in an intelligible fashion and written in standard English?

Reviewer #1: Yes

Reviewer #2: Yes

5. Review Comments to the Author

Response:

Thanks for providing good feedback to the manuscript.

Reviewer #1: Introduction

Query

Language structure issues in “…popular staples in West Africa [3] while serving as important source of dietary calories thus contributing on average, more than 200 Kilocalories per person per day to ….“ the sentence could be better written as “….popular staples in West Africa [3], serving an important source of dietary calories with contribution of 200 Kilocalories on average per person per day to o…”

Line 5 says “… diverse agro-ecologies of the country …”, What country? Nigeria? If West Africa please refer to it as you have not mentioned any country at this point.

Response

We have re-adjusted the entire introduction to make it more suitable for the readers

Methods

Query

The authors have not stated how long (days) the accessions were planted before harvesting

The authors made mention of agronomic practices regularly applied to the crop, however, agronomic practices are too vague and the author should state these agronomic practices in supplementary if it will disrupt the section structure

For the DNA extraction, the authors stated they use young leaves, how are the young leaves selected/identified? (i.e. was it the youngest leaves from the yam or the youngest leaves from a collar, trifoliate, etc.). This is vital to grade uniformity of the harvest tissues because if younger leaves were selected randomly in the yam from accessions to accession, then the data's reproducibility is questioned.

Response

We appreciate feedback from the reviewer, we have provided all missing information in the revised manuscript. 

Results

Query

In Table 3, what does the a, b, c, d … represent? does it mean how statistically different the traits are? Indicate this in the table legend.

Response

These stand for the difference observed among the traits we have adjusted this and provided explanation below table 3.

Query

Figure 1 needs better resolution, the labels are not clearly visible

Figure 2 legend needs to be detailed better

Response

We have provided high quality figure in the revised manuscript 

Query

The heat map section should be given careful attention, the authors need to rearrange the letter in a clear way and specify the most significant. The heat map figures need better resolution as the accession could hardly be seen. The authors may only plot the map of the top most significant variation and include the others/this as supplementary material.

Response

We have removed this from the manuscript to avoid any confusing sentence 

Query

Discussion

The discussion therein contains a detailed explanation of the results and the authors clearly identified the gap, however, the discussion section needs to be well-arranged with enough paragraphs separating different objectives and interrelating results. As it is, the discussion is vague, and if possible should have subheadings.

Response: 

Thanks for your feedback, we have corrected the entire discussion in the revised manuscript.

Query

Conclusion

The conclusion is too minimal and a more robust conclusion highlighting the significant high impact of this work needs to be established. The last sentence for instance can be improved upon.

Response

We have adjusted the conclusion with more informative sentences.

Reviewer #2: The manuscript “Exploring the Genetic Diversity and Population Structure of Aerial yams (Dioscorea bulbifera L.) DArT-seq and agronomic traits” presents insightful knowledge on the diversity among the bulbifera species of yam.

To make the manuscript more presentable the following comments can be taken into consideration:

Query

1. Abstract: Very simplified but can be improved

Response 

We have adjusted the abstract in the revised manuscript 

2. Introduction:

a. The introduction needs some more literature that will cut across some findings and the importance of the use of SNPs markers for diversity studies.

Response

We have adjusted the introduction with genetic diversity related to yam 

Query

b. The production and distribution of the yam specie used for the study should be emphasized with facts.

Response

We have provided details information about the production. However, there is no statistic attached as the crops is generally grown with additional yam with no attention 

Query

c. Check the formatting.

Response: Addressed in the revised manuscript

d. Is there any study on D. bulbifera? Some highlights can be in the introduction.

Response: Currently there is no available genetic study on the D. bulbifera. Some unpublished annual report from the NRCRI, Umudike Nigeria 

Query

e. Other minor typo errors have been indicated in the attachment.

Response: Thanks for the suggestions we have corrected all the errors in the revised manuscript 

3. Methodology:

Query

a. Kindly provide some information on the nature of the soil for the experimental site.

Response: We have provided response in the revised manuscript

Query

b. Table 1: Kindly indicate the accessions that were samples from IITA and NRCRI in the table.

Response: We have provided this information in the revised manuscript

Query

c. Kindly indicate the number of replications used for this design.

Response: We have provided this information in the revised manuscript

Query

d. “A total of 7 qualitative morphological descriptors for yam was used to profile the 94 accessions” This statement can begin for the Phenotyping session. From the statement here, you indicated seven but the parameters in the table are 8. Please clarify and make the necessary adjustments. This may affect the analysis, and this must be clear in the results.

Response: We have adjusted all this in revised manuscript 

Query

e. I suggest tuber parameters should be added to give a very clear diversity in the accessions used.

Response: We appreciate suggestion from the reviewer. However, the main objective of this was to use molecular markers to access the diversity as the phenotypic is subjected to environmental variation. We plan to conduct another field evaluation with more traits for critical observation and those parameters will be added. 

Query

f. Pictures that show clear morphological variation can also be very good to show the difference observed.

Response: We have this documented during this cropping season when doing the trait profiling of the large collection 

Query

g. Table 2: Indicate the period in which the parameters were taken.

Response: We have added the missing information 

4. Results:

Query

a. Table 3: This table does not reflect the number of parameters in Table 2. Please clarify and make the necessary adjustments.

Response: We have adjusted the table 3 in the revised manuscript

Query

b. In the abstract author indicated AMOVA results but that is not reflected in the results. I suggest the AMOVA results are indicated and explained.

Response: We have provided AMOVA table summary in the revised manuscript 

Query

c. Figure 3. Explanations about this figure were not stated in the results. It is important to update that.

Response: We have provided update in the revised manuscript

Query

d. Figures 7 and 8 must be together since they communicate the same information.

Response: Thanks for the suggestion which is well appreciated. In this manuscript we prefer the two figure to stand alone for good clarity and discussion.

Query

e. Please note this well: To improve on the results and analysis. Further analysis can be done by combining the phenotypic and genotypic information in one analysis “dendrogram” to show the relationship and to support the information acquired for both data. That will be very good and explanatory enough. As it stands now, the phenotypic and genotypic information are separated and hanging but it will be good to blend the information from both sides together to make the study rich.

Response: This is very good suggestion and as I have mentioned before we are planning for large profiling and this analysis will be conducted once we have several year data with large phenotypic data set. We have conducted similar work in some of our manuscript and this can be easily be replicated once the additional field data is ready

5. Discussion:

a. The authors have elaborated on the required literature, however there is no discussion on the implication of the morphological observations with the marker’s information identified or vice versa.

b. I suggest the authors explain more about that after combining the analysis of the results as indicated in the results session.

Response

We have adjusted the entire discussion. 

6. PLOS authors have the option to publish the peer review history of their article (what does this mean?). If published, this will include your full peer review and any attached files.

Do you want your identity to be public for this peer review? For information about this choice, including consent withdrawal, please see our Privacy Policy.

Reviewer #1: No

Reviewer #2: No

---

## [Decision Letter · Decision Letter 1]

30 Apr 2024

PONE-D-23-28562R1Exploring the Genetic Diversity and Population Structure of Aerial yams (Dioscorea bulbifera L.) DArT-seq and agronomic traits.PLOS ONE

Dear Dr. Agre,

Thank you for submitting your manuscript to PLOS ONE. After careful consideration, we feel that it has merit but does not fully meet PLOS ONE’s publication criteria as it currently stands. Therefore, we invite you to submit a revised version of the manuscript that addresses the points raised during the review process.

 Please submit your revised manuscript by Jun 14 2024 11:59PM. If you will need more time than this to complete your revisions, please reply to this message or contact the journal office at plosone@plos.org. Please include the following items when submitting your revised manuscript:A rebuttal letter that responds to each point raised by the academic editor and reviewer(s). You should upload this letter as a separate file labeled 'Response to Reviewers'.A marked-up copy of your manuscript that highlights changes made to the original version. You should upload this as a separate file labeled 'Revised Manuscript with Track Changes'.An unmarked version of your revised paper without tracked changes. You should upload this as a separate file labeled 'Manuscript'.If applicable, we recommend that you deposit your laboratory protocols in protocols.io to enhance the reproducibility of your results. Protocols.io assigns your protocol its own identifier (DOI) so that it can be cited independently in the future. For instructions see: https://journals.plos.org/plosone/s/submission-guidelines#loc-laboratory-protocols. Additionally, PLOS ONE offers an option for publishing peer-reviewed Lab Protocol articles, which describe protocols hosted on protocols.io. Read more information on sharing protocols at https://plos.org/protocols?utm_medium=editorial-email&utm_source=authorletters&utm_campaign=protocols.

We look forward to receiving your revised manuscript.

Kind regards,

Timothy Omara, PhD

Academic Editor

PLOS ONE

Journal Requirements:

Reviewers' comments:

Reviewer's Responses to Questions

**Comments to the Author**

1. If the authors have adequately addressed your comments raised in a previous round of review and you feel that this manuscript is now acceptable for publication, you may indicate that here to bypass the “Comments to the Author” section, enter your conflict of interest statement in the “Confidential to Editor” section, and submit your "Accept" recommendation.

Reviewer #2: (No Response)

2. Is the manuscript technically sound, and do the data support the conclusions?

Reviewer #2: Yes

3. Has the statistical analysis been performed appropriately and rigorously? 

Reviewer #2: No

4. Have the authors made all data underlying the findings in their manuscript fully available?

Reviewer #2: Yes

5. Is the manuscript presented in an intelligible fashion and written in standard English?

Reviewer #2: Yes

6. Review Comments to the Author

Reviewer #2: The study on Exploring the Genetic Diversity and Population Structure of Aerial yams (Dioscorea bulbifera L.) DArT-seq and agronomic traits is a very extensive study which is accepted in the yam community as required to explore more genotypes for improvement study. However authors need to address the following to make the study scientifically sound for publishing.

Generally,

• All abbreviation should be checked and should be consistent throughout.

Abstract:

• This session is sound but will need some improvement pointing out some key results that are of significant importance to readers. Generalizing the results is not appropriate. Indicate some values and should be reelevate for the study.

Introduction:

• This session was simple and well written.

Materials and methods:

• Phenotyping: The use of just 7 variables seems not too extensive enough for the population in this study. Is there a reason for that? I suggest authors increase the number of variables by including flowering and some tuber characteristics. Remember that yam ontology has enough variable that can bring out the variation between these genotypes in the study.

• The variables codes were not consistent in the entire study. Authors should ensure that that variable codes are consistent.

• This will call for some figures and tables variable labels to be adjusted.

• Authors should add phenotypic clusters to the analysis of the phenotypic data.

• Genotyping analysis: Authors indicated AMOVA in the results, but the tools used for the AMOVA analysis were not indicated in the analysis section. Authors are expected to indicate that.

Results:

• This session needs more improvement.

• Figures: figures were not well cited in text. Some were not talked about completely and others were misplaced. I suggest authors should make it a point to check the figure very well.

• Clustering for phenotypic traits should be included in the results.

• The PCA Biplot can be improved by showing the variation of the genotypes based on source or origin or agroecological zone. This will make the plot very informative and appreciable.

• Summary genotypic statistics can be improved by indicating the Gene diversity and Call rate and other statistics to make table 5 very informative.

• Results on AMOVA were not presented in the results. A separate paragraph should be developed for that.

• The clustering was not explained well. I suggest a separate paragraph to be allocated for that.

• Note that the method of population structure using STRUCTURE, Clustering and AMOVA are different, and each should complement each other. Having them together does not make your store interesting for the study. I suggest authors provide separate paragraphs for STRUCTURE, AMOVA, CLUSTERING and if possible, include PCoA or PCA and indicate how these methods manage the population structure.

• More importantly, I suggest authors combine the results from phenotypic and genotypic to construct a cluster analysis to show the relation among them. That will be very good for this study.

Discussion:

• The discussion should be improved with any further analysis that is performed in the results section.

7. PLOS authors have the option to publish the peer review history of their article (what does this mean?). If published, this will include your full peer review and any attached files.

Reviewer #2: **Yes: **Emmanuel Amponsah Adjei

---

## [Author Response · Author response to Decision Letter 1]

19 Jun 2024

`We have reoganized the data and re-analyse evething to satify the reviewers comments and suggestions. We have improved all content of the manuscript including the discussion. We have added all types of analysis requested by the reviewers.

---

## [Editor Report · Decision Letter 2]

21 Jun 2024

Exploring the Genetic Diversity and Population Structure of Aerial yams (Dioscorea bulbifera L.) DArT-seq and agronomic traits.

PONE-D-23-28562R2

Dear Dr. Agre,

We’re pleased to inform you that your manuscript has been judged scientifically suitable for publication and will be formally accepted for publication once it meets all outstanding technical requirements.

Kind regards,

Timothy Omara, PhD

Academic Editor

PLOS ONE
---

## [Editor Report · Acceptance letter]

9 Jul 2024

PONE-D-23-28562R2 

PLOS ONE

Dear Dr. Agre, 

I'm pleased to inform you that your manuscript has been deemed suitable for publication in PLOS ONE. Congratulations! Your manuscript is now being handed over to our production team.

Kind regards, 

on behalf of

Dr. Timothy Omara 

Academic Editor

PLOS ONE